# Biomarkers of Exposure and Potential Harm in Two Weeks of Smoking Abstinence: Changes in Biomarkers of Platelet Function, Oxidative Stress, and Inflammation

**DOI:** 10.3390/ijms24076286

**Published:** 2023-03-27

**Authors:** Patrudu Makena, Eric Scott, Peter Chen, Hsiao-Pin Liu, Bobbette A. Jones, Gaddamanugu L. Prasad

**Affiliations:** 1RAI Services Company, 401 N. Main Street, Winston-Salem, NC 27101, USA; 2Prasad Scientific Consulting LLC, Lewisville, NC 27023, USA

**Keywords:** smoking abstinence, biomarkers of potential harm, biomarkers of exposure, 2,3-d-TXB_2_, LTE_4_, neutrophils, WBC

## Abstract

Chronic cigarette smoking is a major risk factor for many serious diseases. While complete cessation of smoking is the best option to reduce harm from smoking, adverse impacts of smoking on health could persist for several years after cessation. Therefore, Biomarkers of Potential Harm (BoPH) are useful in interim evaluations of the beneficial effects of smoking cessation or switching to potentially lower-risk tobacco products. A 14-day smoking abstinence study was conducted under clinical confinement conditions and enrolled 70 subjects into younger (24–34 years, *n* = 33) and older (35–60 years, *n* = 37) age cohorts. Biomarkers of Exposure (BoE), which indicate exposure to nicotine and other toxicants, were measured at baseline, 7 and 14 days. Several BoPH including previously identified eicosanoids (leukotriene 4 (LTE4) and 2,3-dinor thromboxane _2_ (2,3-d-TXB_2_) and others were evaluated. Significant declines in BoE, LTE_4,_ 2,3-d-TXB_2_, neutrophils, WBC and select RBC, and arterial blood gas parameters were observed in both age cohorts at Days 7 and 14 compared to baseline, while other BoPH (e.g., FeNO) showed age-related effects. Rapid and reproducible reductions in LTE_4_, 2,3-d-TXB_2_ WBC, and neutrophil counts were consistently detected following smoking abstinence, indicating the value of these markers as useful BoPH.

## 1. Introduction

Cigarette smoking is a major risk factor for cardiovascular diseases (CVD), cancer, and chronic obstructive pulmonary disease (COPD), which typically develop over several years of sustained smoking [1]. Combustion-related toxicants in cigarette smoke cause oxidative stress and chronic inflammation, among other perturbations, which lead to smoking-related diseases in susceptible smokers. Smoking cessation is the best option for smokers to reduce the risk of developing smoking-related diseases [2].

The Family Smoking Prevention and Tobacco Control Act authorized the Food and Drug Administration (FDA) to regulate the marketing of tobacco products in USA [3]. Premarket and modified risk tobacco product authorizations are two such mechanisms of regulation under the tobacco control act [3]. The FDA established several Harmful and Potential Harmful Constituents (HPHCs) in cigarette smoke and identified them as causative toxicants of smoking-related diseases [4]. Biomarkers are useful tools to assess exposure to HPHCs and the biological effects from tobacco exposure [5].

In the context of tobacco use-related health effects, Biomarkers of Exposure (BoE) and Biomarkers of Potential Harm (BoPH) have been broadly recognized [5,6,7]. BoE quantify exposure to the HPHCs, whereas BoPH inform of the effects of exposure to toxicants. Biomarkers of exposure include nicotine, carbon monoxide and tobacco-specific nitrosamines (NNAL, NNN, NAT, and NAB). Exposure to nicotine is measured as a composite of nicotine and five other metabolites and expressed as total nicotine equivalents. Changes in smoking related BoPH precede the manifestation of the disease. The BoPH are distinct from surrogate biomarkers [8]. BoPH could serve as intermediate endpoints for assessing potential health risks of tobacco use as well as determining how soon physiological functions may improve following smoking cessation [6].

Cigarette smoking impacts multiple organ systems and diverse cell types [1]. For example, smoking-induced oxidative stress activates platelets and renders them prothrombotic. Hyperactive platelets are important mediators of smoking-induced cardiovascular outcomes [1]. Arachidonic acid metabolites, play an important role in platelet activation and atherothrombosis [9]. It is well established that smokers have elevated levels of thromboxane A_2_ metabolites including 11-dehydro-thromboxane B_2_ (11-dh-TXB_2_) and 2, 3, dinor thromboxane B_2_ (2,3-d-TXB_2_), which are widely recognized markers of platelet activation [10,11]. For example, we and others have shown in cross-sectional studies that thromboxane metabolites and other arachidonic acid (AA) metabolite levels are higher in generally healthy smokers [12,13,14]. Further, platelet activation markers have been reported to decrease following smoking cessation, and only two weeks of smoking cessation was reported to improve platelet aggregability [15,16]. Thus, changes in the levels of platelet activation markers could inform of the early biological effects of cigarette smoking or cessation and serve as BoPH.

Other commonly employed BoPH encompass biochemical (e.g., isoprostane F2), cellular (e.g., white blood cell counts [WBC]), and physiological (e.g., forced expiratory volume in one second [FEV1]) markers, among other categories [6,17,18]. Some important attributes of BoPH are their relation to a disease pathway and its responsiveness to smoking cessation. Chronic cigarette smoking profoundly alters human biology [1] and many of those effects are not readily reversed with smoking cessation. For example, FEV1, a key measure of lung function, requires relatively sustained smoking cessation of several months to years to observe a significant improvement [19,20].

In an effort to identify BoPH that rapidly respond to smoking cessation and are mechanistically linked to the development of smoking-related diseases, RAI Services Company (RAIS) has identified two urinary eicosanoids, leukotriene E_4_ (LTE_4_) and 2,3-d-TXB_2_ [14], and shown that the levels of eicosanoids rapidly decline upon smoking cessation or switching to non-combustible tobacco products [21,22]. With an objective to further qualify LTE_4_ and 2,3-d-TXB_2_, we conducted a two-week biomarker study under clinical confinement, which we refer to as the Smoking Abstinence (SAB) study. Given that age and duration of chronic smoking could impact BoPH differentially, we assessed the effect of smokers’ age on biomarker changes upon smoking abstinence by recruiting study participants into two broad age groups. We also measured changes in select BoE to assess declines in exposure to HPHCs following smoking cessation. Further, we explored the effect of two-week smoking cessation on several other cellular (hematological) and physiological BoPH.

## 2. Results

In order to assess biochemical and cellular changes following smoking abstinence and identify potential BoPH that could be utilized for regulatory purposes, we conducted a clinical study, in which smokers were confined to residential conditions for 14 days. Figure 1 summarizes the study design for the SAB clinical study, in which adult smokers abstained from smoking and nicotine-containing products for 14 days in a confinement setting. The SAB was a single-center, two-cohort smoking abstinence study, in which generally healthy adult male and female smokers participated (Table 1). Smokers of 10–20 cigarettes per day for at least five years prior to screening were recruited. Seventy subjects (51 males and 19 females) were enrolled into the study, 33 subjects in the 24–34 age cohort and 37 in the 35–60 age cohort. Sixty-eight subjects completed the study and two subjects discontinued early. Additional details of the study conduct are detailed in the Appendix A.

### 2.1. Urinary Biomarkers of Exposure

We measured exposure to several representative HPHCs at baseline and following smoking abstinence for 7 and 14 days. Bioanalytical methods for urinary BoE are summarized in Appendix A. Declines in BoE upon smoking abstinence were in alignment with the respective reported half-lives, and profound reductions were observed in all urinary BoE on Days 7 and 14 (Table 2). Total nicotine equivalents (NicEq-T), a composite measure of the excretion of nicotine plus five metabolites, declined from baseline on both Days 7 and 14 (about 98% reduction) in both age cohorts. The 2-cyanoethylmercapturic acid (CEMA), a representative BoE for volatile organic compounds [7,23,24] markedly decreased from baseline on Day 7 (88%) and Day 14 (91%) in both age cohorts.

The urinary levels of the tobacco specific nitrosamines (TSNAs) also rapidly declined, reflective of their half-lives. Reductions in NNAL-T (a biomarker for NNK exposure) levels were 71% and 83% on Days 7 and 14, respectively, for both age cohorts. NNN-T levels decreased rapidly in both age cohorts on Days 7 and 14, with a 99% change from the baseline values. Similarly, notable declines in NAB-T and NAT-T levels were observed for both age cohorts on both days.

### 2.2. Blood Biomarkers of Exposure

Bioanalytical methods for measuring blood BoE are summarized in Appendix A. Blood BoE results for plasma nicotine, plasma cotinine, and carboxyhemoglobin (COHb) were significantly lower on Day 7 and Day 14 for both age cohorts (Table 2). The levels of plasma nicotine and cotinine demonstrated a 98% reduction in nicotine levels for both age cohorts on Days 7 and 14. Mean Blood COHb levels for both age cohorts were approximately 60% lower on Days 7 and 14 relatives to baseline values.

### 2.3. Biomarkers of Potential Harm

Previous work has established that smoking abstinence and/or exclusive switching to non-combustible tobacco products, such as moist snuff and ENDS (i.e., Vuse ENDS), resulted in significant declines (24–43%) in the BoPH LTE_4_ and 2,3-d-TXB_2_ levels within 5 days of product switching [22]. In this study, we determined whether continued smoking abstinence would further lower these BoPH and if age would impact them. Bioanalytical methods for measuring BoPH are summarized in Appendix A. Urinary 2,3-d-TXB_2_ levels 7 days after abstinence decreased by approximately 47% and 28% for the younger and older age cohorts, respectively (Figure 2A and Appendix A). Continued abstinence for 14 days sustained the decrease in 2,3-d-TXB_2_ to about 40% for both cohorts, suggesting a reversal of platelet activation (Appendix A).

Urinary levels of LTE_4_ were also reduced after 7 days of smoking abstinence by approximately 42% and 26% compared to baseline for the younger and older cohorts, respectively (Figure 2B; Appendix A). Similar reductions were also observed at 14 days, where LTE_4_ levels decreased approximately 36% and 31% in the younger and older cohorts, respectively. Reduction in LTE_4_ levels suggest a reduced state airway hypersensitivity. While additional statistically significant differences in the mean differences from baseline were found for other BoPH tested, they were not consistent. For example, mean differences for 8-iso-PGF2α from baseline were statistically significantly lower only for the younger cohort, representing a 36% and 29% reduction on Days 7 and 14, respectively (Appendix A). Other BoPH changes were also not significant and not consistent.

### 2.4. Hematological Biomarkers

Cigarette smoking is associated with chronic inflammation, as evident from elevated WBC and leukocyte subtypes [25], which is a common feature of smoking-related diseases [1]. To assess which of the cell types rapidly reverses following smoking abstinence, CBC with differential data was analyzed at baseline and at 7 and 14 days following smoking abstinence (Table 3, Appendix A). By Day 7, the WBC count decreased by 13% and 25% in the younger and older age cohorts, respectively. The declines in WBC counts appear to be rapid, as the Day 14 counts (11% and 21% in the younger and older age cohorts, respectively) were comparable to those at Day 7 of abstinence. Among the major subpopulations of WBC, neutrophil counts were significantly lower after 7 days (18% and 31% decline) and 14 days (17% and 28% decline) of abstinence in both the younger and older age cohorts, respectively, with the neutrophil counts in the older cohort demonstrating larger percent declines compared to the younger cohort. Lymphocyte counts, while lower in both age cohorts, only reached statistical significance in the older age cohort at Day 7 (16%) and Day 14 (11%). Thus, smoking abstinence leads to a rapid decrease in WBC, driven by changes in neutrophil counts, and those declines were steeper in the older age cohort.

Additionally, smoking also results in increases in red blood cell (RBC) counts and other hematological parameters [25]. Smoking abstinence for 7 days resulted in decreased levels of RBC (2% and 3%), hematocrit (2% and 3%), and hemoglobin (3% and 3%) in the younger and older age cohorts, respectively (Table 3). At 14 days of abstinence, further declines in RBC counts, hematocrit, and hemoglobin (4% for all parameters) were observed in both age cohorts were observed.

### 2.5. Physiological Biomarkers of Potential Harm

Long-term cigarette smoking alters pulmonary ventilation, which leads to impaired gas exchange and consequently a decline in lung function. We assessed arterial blood gas (ABG) parameters as a measure of lung function at baseline and 14 days of smoking abstinence (Table 4). Statistically significant changes in the mean differences from baseline for each cohort were found in several of the ABG indices on Day 14 relative to baseline values. For the older cohort, the mean difference of the partial pressure of oxygen on Day 14 from baseline was statistically significant, representing a 4% increase from baseline. The partial pressure of CO_2_ means differences from baseline on Day 14 for the younger cohort and the older cohort increased 3% and 5% from baseline, respectively. The bicarbonate mean differences from baseline on Day 14 for the younger and older cohorts increased 3% and 4% from baseline, respectively. The percent oxygen saturation mean difference from baseline on Day 14 for the older cohort was 1% higher than baseline. Overall, the older age group showed improvement compared to the younger age group. However, for FeNO, a marked increase from baseline was observed for only the younger cohort (reflecting a 56% improvement) on Day 14.

## 3. Discussion

One criterion for the selection of a smoking related BoPH is that the biomarker must be responsive to changes in smoking status. In efforts to identify and qualify BoPH, we evaluated several biomarkers in a 14-day smoking abstinence study. We found that smoking abstinence rapidly restores the levels of select biomarkers of risk to a normal state (i.e., levels comparable to those of non-smokers). We demonstrated that LTE4 (a marker of airway hypersensitivity) and 2,3-d-TXB2 (a marker of platelet aggregation) rapidly declined in generally healthy smokers in response to smoking abstinence.

Several studies have shown that BoE are significantly higher in smokers and decline rapidly upon smoking cessation or with use of noncombustible tobacco products [26,27,28,29]. Although the levels of BoPH differ between asymptomatic smokers, few studies have examined how rapidly these levels decline with smoking cessation. For example, cigarette smoking is known to cause oxidative stress and induce chronic inflammation, which are central mechanisms in the development of CVD, COPD, and cancer [6]. Thus, a reduction in the BoPH indicative of oxidative stress and resolution of chronic inflammation upon cessation (or switching to alternative non-combusted tobacco products) reflects a return to normal physiology and might indicate reduced disease risk compared to continued smoking. Therefore, we focused on a subcategory of BoPH, biomarkers of risk that rapidly respond to changes in smoking status, as potential tools for tobacco product evaluation.

The levels of some eicosanoids are known to increase in smokers, most likely due to smoking-induced oxidative stress, and their levels are reported to decrease upon smoking abstinence. Several studies have shown that smokers exhibit higher urinary levels of the AA metabolites LTE_4_, 8-iso PGF_2α_, 2,3-d-TXB_2_ and 11-dehydrothromboxane B_2_ (11-dh-TXB_2_) compared to non-smokers or users of moist snuff [11,14,30]. In the current study, significant reductions in 8-iso PGF_2α_ levels were observed in only the younger age cohort at both 7 and 14 days of smoking abstinence (36% and 29%, respectively; Appendix A). A recent meta-analysis also reported higher levels of 8-iso PGF_2α_ in smokers and noted the availability of limited information on smoking cessation [30]. Modest declines in 8-iso PGF_2α_ levels were also reported in a 3-month smoking cessation study with limited subjects [31].

Corroborating our previously published work [22], the levels of LTE_4_ in the current study were significantly lower in both age cohorts after 7 days of smoking abstinence, and those decreases were sustained at 14 days. The percent decline in this BoPH is comparable to that observed in non-smokers and moist snuff consumers [14], suggesting that near-maximal reductions could be achieved within a week and are sustained further in smoking abstinence.

Cigarette smoking is a major risk factor for CVD, and smokers exhibit elevated levels of thromboxane A2 (TXA_2_), which promotes platelet activation, a key event in the development of CVD [32,33]. The short-lived TXA_2_ is metabolized to 2,3-d-TXB_2_ and 11-dh-TXB_2_, which are routinely measured in urine as measures of platelet activation. The declines in 2,3-d-TXB_2_ levels in 7 and 14 days are consistent with the findings from our previous work, which showed similar rapid declines following smoking abstinence or switching to ENDS products [21,22]. Several studies have reported that levels of 11-dh-TXB_2_ are also higher in smokers compared to non-smokers (including moist snuff users) [12,13,14]. A meta-analysis [11] confirmed a higher urinary excretion of 11-dh-TXB_2_ in smokers compared to non-smokers, and also discussed several cessation studies that reported a rapid decline in the urinary levels of 11-dh-TXB_2_. For example, 2,3-d-TXB_2_ and 11-dh-TXB_2_ declined by three days in women (*n* = 8) who quit smoking [16]. While, we found reductions in 11-dh-TXB_2_ previously, we were unable to replicate a consistent, statistically significant reversibility of urinary levels of 11-dh-TXB_2_ following smoking abstinence in this or previous studies [22].

Morita eta al [15] reported that smoking cessation reduced agonist-induced platelet aggregation in 2 weeks smoking cessation, which was sustained for 4 weeks in healthy male medical students with a mean age of 27.4 years. A more recent report by Kito et al. [34] indicated that platelet aggregability returns to baseline levels after 12 weeks of smoking cessation, with a transient increase at 4 and 8 weeks; this study enrolled both male and female subjects with ages ranging from 35 to 71. In our study, platelet activation was not statistically significant in both age groups at baseline, 7 and 14 days of smoking abstinence. The variance between the two studies could be due to differences in subject demographics. As discussed, the levels of 2,3-d-TXB_2_ were significantly lower at 7 and 14 days relative to baseline.

Other BoPH that are consistently elevated in smokers include the total WBC count and its composite leukocyte subpopulations, and RBC parameters. The rapid declines we observed in WBC, neutrophil, and lymphocyte counts in smoking abstinence are consistent with the findings from a previous 2-week smoking cessation study [35]. Further, short-term switching to cessation or electrically heated cigarettes also significantly lowers neutrophil count within three days [36].

Chronic cigarette smoking alters pulmonary ventilation and leads to impaired gas exchange, which can be measured via ABG analysis. The impact of age on ABG parameters (PaO_2_, PaCO_2_, and O_2_ saturation) has not previously been clearly described in smokers [37]. However, in our study, markedly lower baseline values of the ABG parameters in the older cohort compared to the younger cohort indicated an impact of prolonged exposure to toxicants present in cigarette smoke (based on years of smoking [Table 1]) and a decline in lung function. The 14 days of smoking abstinence significantly improved PaO_2_ and O_2_ saturation in the older age cohort.

Lower than normal levels of PaCO_2_ at baseline in the older cohort suggest long-term smoking impacts, CO_2_ clearance, and causes an increase in CO_2_ loss. The decrease in PaCO_2_ could be due to the hypoxia that is generally observed in long-term smokers, which stimulates hyperventilation in an attempt to correct hypoxia at the expense of CO_2_ loss [38]. However, this study recruited asymptomatic/generally healthy smokers, and hence the improvement in the ABG parameters is small but statistically significant.

Oxidative stress caused by cigarette smoking has been known to reduce the levels of FeNO in smokers’ lungs [1], and the levels of exhaled nitric oxide are reported to increase following smoking cessation [39,40]. In this study, we report a significant increase in FeNO on Day 14 compared to baseline only in the younger cohort. The impact of smoking on respiratory inflammation depends on cumulative pack-years and time since smoking abstinence. In addition, it is reported that age may impact FeNO levels [41]. Sustained long-term smoking and age augment respiratory inflammation [42]. Our data show that some biomarkers, such as FeNO, require longer than two weeks of smoking cessation in older smokers to observe a prominent change, and age may influence the rapidity of reversal of smoking-induced changes.

Among the strengths of this study, no pharmacological intervention was offered to aid with smoking abstinence, and the subjects tolerated the smoking restriction well, indicating that the clinical risk markers could be useful. Additionally, the BoPH rapidly responded to smoking abstinence in an age-dependent way and are therefore useful for product evaluation in short-term clinical studies. The WBC markers (lymphocytes and neutrophils) and RBC markers (RBC counts, hemoglobin, and hematocrit) were responsive to smoking abstinence (SAB study) in study populations ranging from 21 to 60 years. These findings demonstrate that these markers are responsive to smoking status and are not confounded by age. Since some of the favorable BoPH changes in this short-term SAB study are also replicated in longer-term ambulatory studies using ENDS or heated tobacco products [43,44], short-term evaluation of BoPH allows for rapid evaluation of tobacco products for regulatory purposes.

Some aberrant effects of smoking on normal biological processes persist for an extended period following smoking cessation or switching use to products with modified risk potential. Thus, the 14-day SAB study may not completely capture all changes, or some of the changes could be adaptive and might need to be stabilized. However, our previous cross-sectional study showed the levels of LTE_4_ and 2,3-d-TXB_2_ are about 47% higher in smokers compared to non-smokers [14], which is comparable to the degree of reduction achieved in the SAB study (Figure 2; Appendix A). This indicates that near-maximal recovery of the BoPH was achieved in two weeks in both the younger and older cohorts. Further, declines in WBC counts reported in the SAB study were also noted in longer-term ambulatory studies in which smokers were switched to heated tobacco products [44]. A second limitation of the study is that due to the limited enrollment of women in the study, we were not able to assess the impact of sex as a confounding factor. However, two studies that had a more even distribution of males and females also reported reductions in WBC, neutrophils, and lymphocytes upon smoking cessation for two weeks [35] or 26 weeks [45], although the studies did not specifically analyze the effect of sex.

In summary, we have further characterized LTE_4_ and 2,3-d-TXB_2_, neutrophils, and WBC, and have shown improvements in select RBC parameters in smoking abstinence. Hence, these BoPH are relevant early clinical markers of smoking-related diseases. Taken together with the markedly lower exposure to BoE and the improvements in BoPH, these data suggest some key biological processes start recovering quickly following a short period of smoking abstinence. Further, the BoPH are useful tools to evaluate the effects of smoking cessation and for tobacco product regulation.

## 4. Materials and Methods

### 4.1. Ethical Conduct

The SAB study (ClinicalTrials.gov Identifier NCT04979013) was approved by a fully accredited Institutional Review Board (Advarra; FDA/Office of Human Research Protections IRB#00000971) and was conducted under the principles of the International Council for Harmonisation of Technical Requirements for Pharmaceuticals for Human Use (ICH) Guideline for Good Clinical Practice (GCP) E6(R2). Advarra’s headquarters is at 6100 Merriweather Dr., Suite 600 Columbia, MD.

### 4.2. Study Design

The primary objective of the SAB study was to determine the levels of LTE_4_ and 2,3-d-TXB_2_ in the two age cohorts during two weeks of smoking abstinence. The rationale for biomarker selection and details of bioanalytical assays are summarized in the Appendix A (Appendix A). The primary study endpoints were the BoPH urinary AA metabolites 2,3-d-TXB_2_ and LTE_4_ in samples collected on Day −1 (baseline) and Day 14.

### 4.3. Statistical Analysis

All enrolled subjects were included in the descriptive summary of demographics and baseline characteristics, which are presented by age cohort. A two-sided paired t-test was used to assess the statistical significance of changes from baseline to Day 7 or Day 14. All statistical analyses were performed using SAS (100 SAS Campus Dr, Cary, NC, USA), and statistical significance was considered at *p* ≤ 0.05. For the biomarker endpoints evaluated, only those cohort comparisons that showed at least one statistically significant result and showed consistent patterns are discussed.

## Figures and Tables

**Figure 1 ijms-24-06286-f001:**
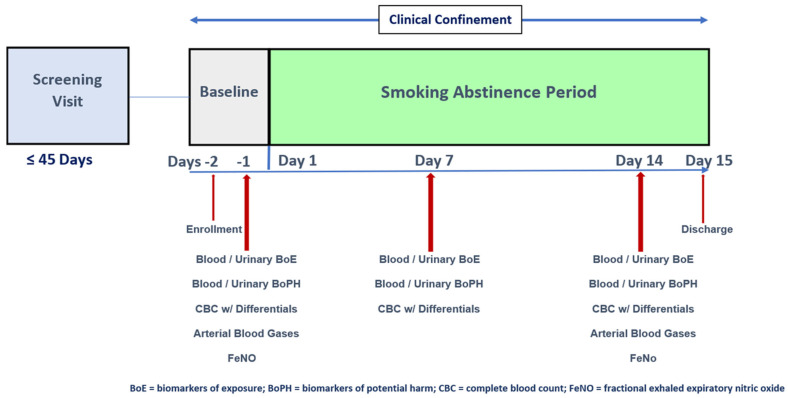
Study Design of Smoking Abstinence Study. The smoking abstinence study consisted of the 14-day clinical confinement of smokers to evaluate biomarker changes. Blood and urine samples were collected at baseline (Day -1) and Days 7 and 14 of smoking abstinence. Biomarker measurements were made at baseline after 7 days and/or 14 days of smoking abstinence, as indicated.

**Figure 2 ijms-24-06286-f002:**
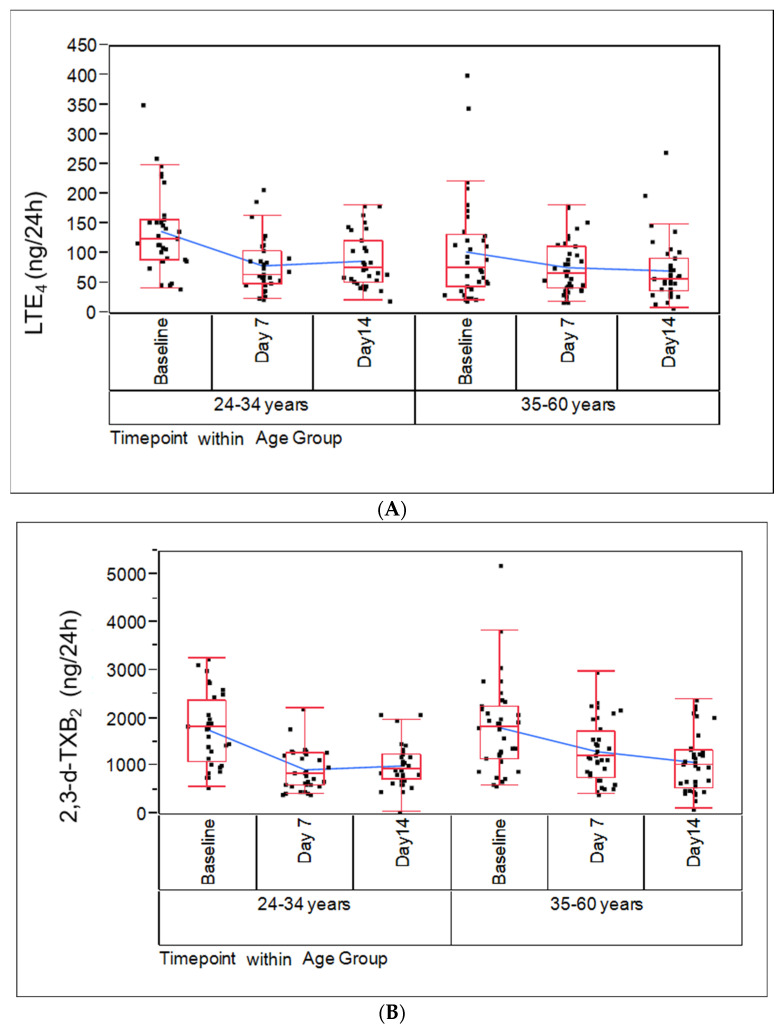
Change in leukotriene 4 (**A**) and 2,3-dinor-thromboxane B2 (**B**) levels following smoking abstinence. Leukotriene 4 (**A**) and 2,3-dinor-thromboxane B2 (**B**) levels rapidly decline following smoking abstinence. Urine samples from a 24 h period obtained at baseline Day 7 and Day 14 from smokers switched from usual brand (UB) cigarettes to smoking abstinence were used to measure leukotriene 4 (**A**) and 2,3-dinor-thromboxane B2 (**B**) by LC/MS-MS. Each individual data point represents subject-level measurements. *p* < 0.01 when comparing baseline with post-switching to abstinence (Day 7 and Day 14) for both Leukotriene 4 and 2,3-dinor-thromboxane B.

**Table 1 ijms-24-06286-t001:** Demographics and Subject Characteristics of Smoking Abstinence Study.

		Age Cohort		
		24–34 (N = 33)	35–60 (N = 37)	Study Overall (N = 70)
Sex n (%)	Female	4 (12%)	15 (41%)	19 (27%)
	Male	29 (88%)	22 (59%)	51 (73%)
Race n (%)	Black or African American	5 (15%)	3 (8%)	8 (11%)
	White	26 (79%)	31 (84%)	57 (81%)
	Multiple	2 (6%)	3 (8%)	5 (7%)
Ethnicity n (%)	Hispanic or Latino	3 (9%)	0 (0%)	3 (4%)
	Not Hispanic or Latino	30 (91%)	37 (100%)	67 (96%)
Age ^a^ (years)	Mean (SD)	30.0 (3.12)	49.2 (8.10)	40.2 (11.52)
FTND Score	Mean (SD)	5.5 (1.70)	6.0 (1.57)	5.8 (1.63)
Years of Smoking ^b^		12.8 (4.08)	29.3 (11.30)	21.5 (11.94)
Cigarettes Smoked per Day		17.6 (4.48)	17.4 (4.02)	17.5 (4.21)
Cigarette Variety n (%)	Menthol	10 (30%)	7 (19%)	17 (24%)
	Non-Menthol	23 (70%)	30 (81%)	53 (76%)

^a^ Age is calculated as the age at date of informed consent. ^b^ Years of Smoking is calculated as informed consent date minus smoking start date. FTND = Fagerström Test for Nicotine Dependence (0 = no dependency; 1 to 2 = low dependency; 3 to 4 = low to moderate dependency; 5 to 7 = moderate to high dependency; 8 to 10 = high dependency); n = number of subjects within a category; N = total number of subjects; SD = standard deviation.

**Table 2 ijms-24-06286-t002:** Summary of Changes in Urinary and Blood Biomarkers of Exposure on Days −1, 7, and 14 of Smoking Abstinence.

Urine Biomarkers			Age Cohort	
Biomarker (Units)	Statistics	Time Point	24–34 Years	35–60 Years
NicEq-T (mg/24 h)	Mean ± SD (*n*)	Day −1	19.4 ± 6.6 (32)	17.3 ± 6.4 (37)
		Day 7	0.4 ± 0.2 (32)	0.4 ± 0.1 (37)
		Day 14	0.4 ± 0.3 (32)	0.4 ± 0.2 (36)
	Percent Change	Day 7 vs. Day −1	−98% *	−97% *
		Day 14 vs. Day −1	−98% *	−98% *
CEMA (µg/24 h)	Mean ± SD (*n*)	Day −1	273.6 ± 115.1 (32)	240.3 ± 91.1 (37)
		Day 7	32.0 ± 12.8 (32)	28.1 ± 14.6 (37)
		Day 14	24.3 ± 9.3 (32)	21.4 ± 13.1 (36)
	Percent Change	Day 7 vs. Day −1	−88% *	−88% *
		Day 14 vs. Day −1	−91% *	−91% *
NNN-T (pg/24 h)	Mean ± SD (*n*)	Day −1	16,380 ± 10,164 (32)	20,380 ± 39,703 (37)
		Day 7	240.0 ± 101.3 (32)	251.9 ± 80.8 (37)
		Day 14	232.9 ± 146.7 (32)	232.7 ± 124.6 (36)
	Percent Change	Day 7 vs. Day −1	−99% *	−99% *
		Day 14 vs. Day −1	−99% *	−99% *
NAB-T (ng/24 h)	Mean ± SD (*n*)	Day −1	66.4 ± 40.4 (32)	55.7 ± 35.3 (37)
		Day 7	2.4 ± 1.0 (32)	2.5 ± 0.8 (37)
		Day 14	2.3 ± 1.5 (32)	2.3 ± 1.2 (36)
	Percent Change	Day 7 vs. Day −1	−96% *	−96% *
		Day 14 vs. Day −1	−96% *	−96% *
NAT-T (ng/24 h)	Mean ± SD (*n*)	Day −1	480.4 ± 292.0 (32)	377.0 ± 226.9 (37)
		Day 7	6.0 ± 2.5 (32)	6.2 ± 2.1 (37)
		Day 14	5.8 ± 3.7 (32)	5.8 ± 3.1 (36)
	Percent Change	Day 7 vs. Day −1	−99% *	−98% *
		Day 14 vs. Day −1	−99% *	−98% *
NNAL-T (ng/24 h)	Mean ± SD (*n*)	Day −1	465.3 ± 249.1 (32)	475.3 ± 263.9 (37)
		Day 7	132.7 ± 85.7 (32)	135.8 ± 84.7 (37)
		Day 14	84.5 ± 47.7 (32)	81.3 ± 54.4 (36)
	Percent Change	Day 7 vs. Day −1	−71% *	−71% *
		Day 14 vs. Day −1	−82% *	−83% *
**Blood Biomarkers**				
Blood CoHB (%)	Mean ± SD (*n*)	Day −1	3.7 ± 1.2 (32)	3.8 ± 1.3 (37)
		Day 7	1.4 ± 0.4 (32)	1.3 ± 0.3 (37)
		Day 14	1.4 ± 0.3 (32)	1.3 ± 0.3 (36)
	Percent Change	Day 7 vs. Day −1	−62% *	−66% *
		Day 14 vs. Day −1	−61% *	−65% *
Plasma Nicotine ** (ng/mL)	Mean ± SD (*n*)	Day −1	5.2 ± 6.5 (32)	4.6 ± 5.1 (37)
		Day 7	0.1 ± 0.0 (32)	0.1 ± 0.0 (37)
		Day 14	0.1 ± 0.0 (32)	0.1 ± 0.0 (36)
	Percent Change	Day 7 vs. Day −1	−98%	−98%
		Day 14 vs. Day −1	−98%	−98%
Plasma Cotinine (ng/mL)	Mean ± SD (*n*)	Day −1	236.6 ± 118.9 (32)	234.2 ± 112.7 (37)
		Day 7	2.9 ± 6.7 (32)	2.1 ± 3.0 (37)
		Day 14	0.6 ± 0.4 (32)	0.5 ± 0.1 (36)
	Percent Change	Day 7 vs. Day −1	−99% *	−99% *
		Day 14 vs. Day −1	−100% *	−100% *

Percent change = (daily mean − baseline mean)/baseline mean × 100, where baseline is Day −1. * Statistical significance of the mean difference from baseline (*p* < 0.05). ** Statistical analysis could not be performed for plasma nicotine as all of the Day 7 and Day 14 data were reported as being below the limit of quantitation. Abbreviations: CEMA: 2-cyanoethylmercapturic acid; CoHB: carboxyhemoglobin; mg: milligram; mL: milliliter; n: sample size; NAB-T: Total N’-nitrosoanabasine; NAT-T: Total N’-nitrosoanatabine; NicEq-T: Total nicotine equivalents; ng: nanogram; NNAL-T: Total 4-(methylnitrosamino)-1-(3-pyridyl)-1-butanol; NNN-T: Total N’-nitrosonornicotine; pg: picogram; SD: standard deviation; µg: microgram.

**Table 3 ijms-24-06286-t003:** Hematological BoPH changes upon smoking abstinence.

Hematological Biomarkers			Age Cohort	
Biomarker (Units)	Statistics	Time Point	24–34 Years	35–60 Years
White blood cells	Mean ± SD (*n*)	Day −2	7.83 ± 1.45 (32)	8.54 ± 2.82 (37)
(10^9^/L)		Day 7	6.81 ± 1.38 (32)	6.41 ± 1.74 (37)
		Day 14	6.95 ± 1.46 (32)	6.71 ± 1.89 (36)
	Percent Change (*p*-value *)	Day 7 vs. Day −2	−13% (<0.0014)	−25% (<0.0001)
		Day 14 vs. Day −2	−11% (0.0077)	−22% (<0.0001)
Neutrophils	Mean ± SD (*n*)	Day −2	4.72 ± 1.24 (32)	5.15 ± 2.27 (36)
(10^9^/L)		Day 7	3.87 ± 0.98 (32)	3.56 ± 1.33 (36)
		Day 14	3.93 ± 0.93 (32)	3.71 ± 1.46 (36)
	Percent Change (*p*-value *)	Day 7 vs. Day −2	−18% (<0.0001)	−31% (<0.0001)
		Day 14 vs. Day −2	−17% (0.0004)	−28% (<0.0001)
Lymphocytes	Mean ± SD (*n*)	Day −2	2.23 ± 0.63 (32)	2.49 ± 0.67 (36)
(10^9^/L)		Day 7	2.13 ± 0.51 (32)	2.09 ± 0.6 (36)
		Day 14	2.20 ± 0.61 (32)	2.23 ± 0.53 (36)
	Percent Change (*p*-value *)	Day 7 vs. Day −2	−5% (0.1201)	−16% (<0.0001)
		Day 14 vs. Day −2	−1% (0.6331)	−11% (0.003)
Red blood cells	Mean ± SD (*n*)	Day −2	5.23 ± 0.37 (32)	4.87 ± 0.43 (37)
(10^12^/L)		Day 7	5.11 ± 0.48 (32)	4.75 ± 0.50 (37)
		Day 14	5.04 ± 0.41 (32)	4.65 ± 0.52 (36)
	Percent Change (*p*-value *)	Day 7 vs. Day −2	−2% (0.0147)	−3% (0.0038)
		Day 14 vs. Day −2	−4% (<0.0001)	−5% (<0.0001)
Hematocrit (%)	Mean ± SD (*n*)	Day −2	46.91 ± 2.68 (32)	44.02 ± 3.56 (36)
		Day 7	45.75 ± 3.39 (32)	42.79 ± 3.97 (36)
		Day 14	45.07 ± 3.04 (32)	42.12 ± 4.29 (36)
	Percent Change (*p*-value *)	Day 7 vs. Day −2	−2% (0.0163)	−3% (0.0007)
		Day 14 vs. Day −2	−4% (<0.0001)	−4% (<0.0001)
Hemoglobin (g/dL)	Mean ± SD (*n*)	Day −2	15.80 ± 0.96 (32)	14.66 ± 1.41 (36)
		Day 7	15.39 ± 1.11 (32)	14.21 ± 1.48 (36)
		Day 14	15.20± 1.06 (32)	14.01 ± 1.57 (36)
	Percent Change (*p*-value *)	Day 7 vs. Day −2	−3% (0.0117)	−3% (0.0003)
		Day 14 vs. Day −2	−4% (<0.0001)	−4% (<0.0001)

* The *p*-value indicates the statistical significance of the comparison between original measurements.

**Table 4 ijms-24-06286-t004:** Changes in arterial blood gases and exhaled nitric oxide upon smoking abstinence.

Physiological Biomarkers			Age Cohort	
Biomarker (Units)	Statistics	Time Point	24–34 Years	35–60 Years
PAO_2_ (mmHg)	Mean ± SD (*n*)	Day −1	89.65 ± 9.26 (32)	81.51 ± 11.02 (37)
		Day 14	91.00 ± 10.02 (32)	85.13 ± 8.17 (36)
	Percent Change	Day 14 vs. Day −1	1%	4% *
PACO_2_ (mmHg)	Mean ± SD (*n*)	Day −1	40.13 ± 3.28 (32)	36.95 ± 3.1 (37)
		Day 14	41.41 ± 2.98 (32)	38.91 ± 3.51 (36)
	Percent Change	Day 14 vs. Day −1	3% *	5% *
O_2_ Saturation (%)	Mean ± SD (*n*)	Day −1	96.71 ± 0.85 (32)	95.67 ± 1.68 (37)
		Day 14	96.68 ± 1.3 (32)	96.38 ± 1.10 (36)
	Percent Change	Day 14 vs. Day −1	0%	1% *
Bicarbonate (mmol/L)	Mean ± SD (*n*)	Day −1	24.75 ± 1.87 (32)	23.44 ± 1.76 (37)
		Day 14	25.38 ± 1.75 (32)	24.28 ± 1.94 (36)
	Percent Change	Day 14 vs. Day −1	3% *	4% *
FeNO (ppb)	Mean ± SD (*n*)	Day −1	12.29 ± 9.07 (31)	15.03 ± 19.56 (33)
		Day 14	19.13 ± 18.42 (31)	15.71 ± 10.78 (35)
	Percent Change	Day 14 vs. Day −1	56% *	5%

Percent change = (daily mean − baseline mean)/baseline mean × 100, where baseline is Day −1. * Statistical significance of the mean difference from baseline (*p* < 0.05). FeNO = fractional exhaled expiratory nitric oxide; O_2_ = oxygen; PAO_2_ = partial pressure oxygen; PACO_2_ = partial pressure carbon dioxide; ppb = parts per billion.

## Data Availability

The data presented in this study are available on request from the corresponding author. The data are not publicly available due to proprietary reasons.

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
