# Peer review of "Biomarkers of Exposure and Potential Harm in Two Weeks of Smoking Abstinence: Changes in Biomarkers of Platelet Function, Oxidative Stress, and Inflammation"

_ijms, 2023, doi:10.3390/ijms24076286_

Round 1

Reviewer 1 Report

Reviewer comments and suggestions

The author in this study studied the biomarkers of Potential Harm (BoPH) which are useful in interim evaluations of the beneficial effects of smoking cessation or switching to potentially reduced risk tobacco products. The methodology included a 14-day smoking abstinence study was conducted and enrolled a total of 70 subjects younger and older participants. Biomarkers of Exposure (BoE) that indicate exposure to nicotine and other toxicants were measured at baseline, 7 and 14 days. The study resulted into a significant declines in BoE, LTE4, 2,3-d-TXB2, neutrophils, WBC and select RBC and arterial blood gas parameters were observed in both age cohorts. Finally, the authors concluded that reductions in LTE4, 2,3-d-TXB2, WBC, and neutrophil counts are consistently detected following smoking abstinence which could be used as a biomarker of potential harm.

Overall, the manuscript was well written. However, a few concerns/comments needed to be explained/modified. 

  1. Line 33 The reference denoted should be a square bracket based on MDPI guidelines.
  2. Line 61 is this correct including “11 -dh -thromboxane B 2”
  3. Line 65 What are healthy smokers here?
  4. Line 69-70 Similar examples need to understand the Biomarker of exposure
  5. Line 86 Why only two weeks? what would be effects after one month of abstinence.
  6. Line 214-215 Please explore the sentence for clarity.
  7. Line 242 How the authors describe these result with 14 days
  8. Line 257 needs more references to add up. “Many studies have reported that”
  9. Line 301 What does it indicate please describe some introductory lines
  10. All references formats should be modified based on MDPI guidelines

Reviewer 2 Report

The manuscript explores the useful Biomarkers of Potential Harm (BoPH) in evaluations of the beneficial effects of smoking cessation or switching to potentially reduced-risk tobacco products. A 14-day smoking abstinence study was conducted and 68 subjects by age cohorts were included. Significant declines in BoE, LTE4, 2,3-d-TXB2, neutrophils, WBC, and select RBC and arterial blood gas parameters were observed on Days 7 and 14 compared to baseline. The study is meaningful, but the sample size is small, and the tracking period is limited (14 days). My questions and comments were attached.
